# An Improved End-to-End Autoencoder Based on Reinforcement Learning by Using Decision Tree for Optical Transceivers

**DOI:** 10.3390/mi13010031

**Published:** 2021-12-27

**Authors:** Qianwu Zhang, Zicong Wang, Shuaihang Duan, Bingyao Cao, Yating Wu, Jian Chen, Hongbo Zhang, Min Wang

**Affiliations:** 1Key Laboratory of Specialty Fiber Optics and Optical Access Networks, Joint International Research Laboratory of Specialty Fiber Optics and Advanced Communication, Shanghai University, Shanghai 200444, China; zhangqianwu@shu.edu.cn (Q.Z.); 15921293764@163.com (Z.W.); yizhi2019@shu.edu.cn (S.D.); ytwu@shu.edu.cn (Y.W.); chenjian@shu.edu.cn (J.C.); wangmin@shu.edu.cn (M.W.); 2College of Communication Engineering (College Microelectronics), Chengdu University of Information Technology, Chengdu 610225, China; zhanghb@cuit.edu.cn

**Keywords:** deep learning, machine learning, Adaboost algorithm, neural networks, optical fiber communication

## Abstract

In this paper, an improved end-to-end autoencoder based on reinforcement learning by using Decision Tree for optical transceivers is proposed and experimentally demonstrated. Transmitters and receivers are considered as an asymmetrical autoencoder combining a deep neural network and the Adaboost algorithm. Experimental results show that 48 Gb/s with 7% hard-decision forward error correction (HD-FEC) threshold under 65 km standard single mode fiber (SSMF) is achieved with proposed scheme. Moreover, we further experimentally study the Tree depth and the number of Decision Tree, which are the two main factors affecting the bit error rate performance. Experimental research afterwards showed that the effect from the number of Decision Tree as 30 on bit error rate (BER) flattens out under 48 Gb/s for the fiber range from 25 km and 75 km SSMF, and the influence of Tree depth on BER appears to be a gentle point when Tree Depth is 5, which is defined as the optimal depth point for aforementioned fiber range. Compared to the autoencoder based on a Fully-Connected Neural Network, our algorithm uses addition operations instead of multiplication operations, which can reduce computational complexity from 10^8^ to 10^7^ in multiplication and 10^6^ to 10^8^ in addition on the training phase.

## 1. Introduction

The application of machine learning technique in optical communication systems has been studied in many fields in recent years [1,2]. In the field of optical communication systems, many parts of the system, such as performance monitoring, fiber nonlinearity mitigation, carrier recovery, and equalization, have been optimized by machine learning and a neural network [3,4,5,6]. In particular, as we all know, chromatic dispersion (CD) and nonlinear Kerr effects in the fiber are the main constraint in the improvement of the signal rate in the optical communication system today [7]. In order to deal with the influence of a nonlinear effect and dispersion on the signal, the traditional method is to add a backward equalizer to reduce the influence of pulse broadening on the bit error rate of the signal. In recent years, as a new method, the artificial neural network (ANN) has been of great interest on channel equalization in the field of wireless communication [8,9,10,11], which shows its advantage on the better bit error rate (BER). Since Deep learning relies on features of the data and situation, it cannot be efficiently trained under the changeable situation of long-distance communication.

An end-to-end communication system, which uses two neural networks (NNs) as transmitter and receiver to be trained for a specific-designed channel model, was presented in Reference [10]. The communication system is regarded as an autoencoder (AE) [12], which is composed of symmetrical NNs modeled as the transmitter and receiver, and intermediate channel modeled as a fixed layer. AE is generally optimized by gradient-based algorithms (stochastic gradient descent [12], Adam algorithm [13], etc.) in the condition of the differentiable loss function (originated from the difference between actual value and label). The AE was firstly applied in the wireless communication [11] to serve as the transmitter and receiver, and it was experimentally demonstrated that it has better performance. The concept of AE has been recently employed to other refined fields, such as orthogonal frequency-division multiplexing (OFDM) [5] and multiple-input multiple-output (MIMO) [14]. In optical fiber communication, the AE has been implemented in the dispersive linear fiber channel [15,16], or in the nonlinear dispersive fiber channel, with nonlinear frequency division multiplexing (NFDM) [17,18] for transmission.

In other fields, the AE is popular in geometric constellation shaping (GCS) in order to obtain the theoretically achievable channel capacity, which appears in a uniformly distributed signal (such as conventional quadrature amplitude modulation (QAM)). Geometric constellation shaping is the process of redesigning the distribution of the constellation points on the I/Q plane. The main purpose of GCS is to achieve balance between Euclidean distance among different constellation points and the energy distribution of the constellation point on the given channel. In References [7,19,20,21], it was demonstrated that the AE can be applied on various optical fiber channel models (such as nonlinear fiber channel [19,20], non-dispersive channel [7], and linear link by QAM [21]) to learn and achieve better geometric constellation.

In the field of AE, generative adversarial networks (GANs) are a new trend to model the non-differentiable or relatively complex channel as a simple differentiable channel on the optical fiber communication [22]. The GANs are simply used to be trained the AE for non-coherent optical fiber communication. Due to additional experiment steps caused by GANs, new data needs to be obtained from the non-differentiable channel model and used to train the AE for better discrimination between synthetic data and original data. Obviously, for ensuring the performance of the GAN, the scale of all training samples (both from non-differentiable channel model and differentiable channel) must be large enough to train GANs to reach the level of relatively accurate channel approximation. Thus, it has the obvious disadvantage in that the less complex non-differentiable channel still takes too much time, and the result is an approximation, which is still different from the actual channel.

In refs. [8,23], a primitive AE model without a channel is proposed. Its purpose is to change the backward propagation from joint optimization to separate optimization. The training method is based on separating autoencoder from a whole to transmitter and receiver apart, using reinforcement learning to train the transmitter, and supervised learning to train the receiver. Compared with joint optimization, separate optimization often leads to training convergence by more training samples [24] due to non-simultaneous optimization.

As an introduction to the development of AE, the paper mentions the corresponding development of GAN, but GAN is not the solution to reduce the computation complexity. Most of the GANs are committed to be trained from fiber channels that require high precision and low complexity [25]. Model collapse is the main problem in GAN. Model collapse comes from not considering the diversity of the data. GAN model training affects whether the final result converges, so different loss functions and condition vector designs are needed for different channels to solve the problem of modal collapse.

In this paper, compared to Reference [15], we implement the improved end-to-end optical fiber system based on reinforcement learning and supervised learning. We realized the limitations of the (Feed Forward Neural Network) FFNN design for communication with blocks over nonlinear channels and improve end-to-end optical fiber system by implementing a reinforcement learning algorithm as a receiver, without the single block but the upsampling block, to receive every message.

We designed the system to train an autoencoder by simulation channel jointly, and optical fiber channel separately. In consideration of the complexity and BER, we broke away from the field of supervised learning and focused on reinforcement learning. Reinforcement learning is a reward guiding behavior that agents learn in the way of “trial and error” and interact with the environment. The goal is to maximize the reward for agents. Compared with supervised learning, reinforcement learning is more sensitive to the training environment and is more conducive to serving as a receiver to distinguish different information.

As a representative of Reinforcement Learning, the Adaboost algorithm [26] has been demonstrated as an addition with many weighted weak classifiers. Its advantage is low complexity and easy training. In addition, multiple weak classifiers include various signal misjudgments caused by pulse broadening, which further reduces the bit error rate through training. In ref. [12], this paper aims to find that AdaBoost has a relatively good effect on classification. Due to training separately, the receiver based on the Adaboost algorithm still has good performance received from an equally-trained transmitter under a different-distance optical fiber channel. We use the Adaboost algorithm as the receiver to design an end-to-end optimized fiber-optic system receiver to overcome inter-symbol interference (ISI) present in Intensity Modulation/Direct Detection (IM/DD) over channels with CD. Since we receive all the block (carry the same message) to solve the same message, pulse broadening resulted by CD between the block cannot influence the message any longer. Based on our previous research [27], we further analyzed the advantages and disadvantages of the Adaboost algorithm, other Machine Learning algorithms, and ANN, and we performed a more detailed analysis of the experimental details, with more comparison. Experimental results show that the receiver with the Adaboost algorithm can outperform the previous end-to-end FFNN and other Machine Learning algorithms in complexity and efficiency. Our results show that the Adaboost algorithm outperforms the reference system in complexity, achieving information rate of 48 Gb/s below the adopted 3.8 × 10^−3^ HD-FEC threshold at the fiber range of 45 km SSMF. The Adaboost algorithm obviously has stronger robustness in the long distance than ANN. Instead, in contrast to ANN, it has less training and running time.

## 2. Principe of Proposed End-to-End Optical Fiber System with the Adaboost Algorithm Receiver Design

### 2.1. The Design of End-to-End Optical Fiber System with the Adaboost Algorithm Receiver

Figure 1 shows a schematic diagram of the entire end-to-end optical communication system. The characteristic of fiber dispersion is that it will lead to the ISI effect from the front and back symbols to the current symbol. We consider optimization of the end-to-end design by replacing ANN with the Adaboost algorithm on the receiver to reduce the computation complexity, at the cost of sacrificing BER performance of the whole optical fiber system (detailed analysis in Section 3). The ANN at the transmitter encodes the message m∈{1,…,m}. Binary Random Sequence generated by the Python Numpy Library [28] is a bit stream consisting of only a unipolar signal (0,1). Therefore, a few bits from this bit stream represent the message m and are encoded into a one-hot vector. Thus, the message m is expressed as a one-hot vector (containing “1” at position m and zero at other positions), and then processed by ANN to obtain n samples to form a waveform. The obtained waveform is upsampled for each sequence point by N times, and then transmitted to a low-pass filter (LPF) and the fiber channel.

As shown in Figure 1, the waveform through the fiber channel is processed by the receiver. ANN and the Adaboost algorithm can also decode the generated waveform into the one-hot vector as in the transmitter before.

In ANN (Figure 1a), the ANN structure of the receiver is reversed with the ANN structure of transmitter. In the Adaboost algorithm (Figure 1b), it is composed of several Decision Trees, and its weight is updated by iterative training. Due to different training methods, ANN can be trained together, and the Adaboost algorithm cannot be trained together. Therefore, we need to train the Adaboost algorithm after training ANN in order to obtain the ANN transmitter. This is detailed description in Section 3.1.

### 2.2. The Design of ANN Transmitter

At the transmitter, as shown in Figure 1, we just make a simple optimization on the transmitter by using ANN based on data structure alone, compared to Reference [17]. First of all, its complexity is lower than parallel structure, making it low-cost and high-speed implementation possible for subsequent online operations. (The parallel structure mentioned here means the parallelism between neural networks.) Secondly, compared to Reference [17], when processing a single module, it reduces redundant parallel processing operations and reduces the calculation space, which is replaced by subsequent upsampling. This method is to reduce the training volume and strengthen relevance in the same block, instead of dealing with the same information through the loose connection of multiple blocks. Each block encodes an independent message m∈{1,…,m} in a group of M messages into a vector of n transmitted samples, thereby forming a symbol waveform. Each message represents the equivalent of $log_{2}M$ bits.

The encoding method is as follows: The message m is encoded into a one-hot vector (the *m*-th element is equal to 1, and the other elements are 0) of size *M*, denoted as $L_{m}\in R^{M}$. The one-hot vector coding is the standard method for representing classification values in most machine learning algorithms. The advantage is that it has a high degree of compatibility with the subsequent calculation probability of the decision-making layer SoftMax.

Considering simplification of the complexity of the transmitter, we incorporate the power normalization into the neural network and use the neural network to process the power normalization while learning.

Usually, a fully-connected K-layer FFNN maps an input vector (one-hot vector) $s_{0}$ to an output vector $s_{k}=f_{ANN}\left(s_{0}\right)$ through the K-layer ANN in the transmitter. Every layer can be described as follows:(1)$$S_{k}=\alpha_{k}\left(W_{k}s_{k-1}+b_{k}\right),\;k=1,\;\ldots ,\;K$$
where $s_{k-1}\in \;R^{N_{k-1}}$ is the output of the (*k* − 1)th layer, which size is $N_{k-1}$ according to the (*k* − 1)th layer weight matrix $W_{k-1}$ and the bias vector of the (*k* − 1)th layer. $\alpha_{k}$ is its activation function. Only two variables $W_{k}$ and $b_{k}$ can be trained. Instead, the activation function $\alpha_{k}$ is to judge whether or not the data point has the feature to be extracted. A one-hot vector is fed to the first hidden layer of the network, and its weight matrix and deviation vector are $W_{1}\in \;R^{M\times 2M}$ and $b_{1}\in \;R^{2M}$, respectively. The second hidden layer has parameters $W_{2}\in \;R^{2M\times 2M}$ and $b_{2}\in \;R^{2M}$. The rectified linear unit (ReLU) activation function is applied to two hidden layers, which acts on each of its input vector elements compared to 0. If its value is lower than 0, it is regarded as having none of the feature value. If the training set is needed to be extracted from the features of negative elements, the leaky ReLU activation function is often added. ReLU can only easily delete inactivated neuron.

The final layer prepares the symbol waveform for transmission. Its parameters are $W_{3}\in \;R^{2M\times n}$ and $b_{3}\in \;R^{n}$, where *n* represents the number of waveform samples of the message. Due to the Mach-Zehnder modulator (MZM) in the IM/DD system, the waveform needs to be converted from a unipolar signal to a bipolar signal to ensure that the linear space in the MZM working range is fully utilized. The ANN transmitter must limit its output value to the relatively linear operating region of MZM [−π/4, π/4], but, in the experiment, it is usually set at a relatively suitable position on [0, π/2]. We achieve this by using a defined clipping activation function for the last layer of output, which combines the following two ReLUs:(2)$$\alpha_{clipping}=\alpha_{\mathrm{ReLU}}\left(x+\frac{\pi }{4}\pm \varepsilon \right)-\alpha_{\mathrm{ReLU}}\left(x-\frac{\pi }{4}\pm \varepsilon \right),$$
where the term ε is defined as the quantization noise caused by DAC. In the simulation, ε is equal to $\sigma_{r}^{2}$. The dimension of this layer determines the oversampling rate of the transmitted signal. Based on the limitation of laboratory instruments, the sampling rate of AWGN transmitted signal must be between 82 Gsa/s and 92 Gsa/s. If we choose the code rate of 12 Gbit/symbol, we need 7 times upsampling to ensure that it is within this range.

So, we have considered 7 times upsampling here in order to match the Arbitrary Waveform Generator (AWG) sample rate. Since fiber dispersion will lead to generating memory between several consecutive symbols, it is the pulse stretching of the waveform from the oscilloscope, and the symbol memory before and after is offset by the operation of upsampling. Therefore, the actual data block is n times the output of the neural network, and the benefits will be reflected in the subsequent analysis of the experimental results. The whole system can be regarded as an autoencoder with effective information rate $R=\log_{2}M$ bits/symbol.

Normally, the training of the neural network can be performed as a supervised training through the contact between the input vector $s_{0}$ and desired output $s_{k}$. Loss function always becomes the bridge through every training model. Considering its function, we use cross-entropy as our loss function, which can be described as:(3)$$L\left(\theta \right)=\frac{1}{\left|S\right|}\sum_{\left(s_{0,i},s_{k,i}\right)\epsilon S}\{\ell \left(f_{ANN}\left(s_{0},i\right),s_{k,i}\right)+\ell \left(1-f_{ANN}\left(s_{0},i\right),1-s_{k,i}\right)\}.$$
$s_{0,i}$, $s_{k,i}$ is the input vector and K-layer ANN output, and |s| is the mini-batch. In the neural network, by training with only one block, backpropagation can easily fall into a local optimum which is harmful to obtaining the optimal classification function. Therefore, a series of blocks is often used as a mini-batch to be put into the neural network for training, and the number of batches is equal to the number of back propagation ℓ(x,y), meaning the cross-entropy, defined as:(4)$$\;\ell \left(x_{i},y_{i}\right)=-\sum_{i}x_{i}\log\left(y_{i}\right).$$
$y_{i}$,$\;x_{i}$ define the I elements of the input vector and the I elements of the output vector. When calculating the cross-entropy, using two functions that are opposite to each other is to increase sensitivity to the feature (always regarded 1 as max). Theoretically, if the output and input is 1 and 0, the value of cross-entropy cannot be a desired huge value to be minimized. So, it needs two functions to describe its loss.

A common method for optimization of the loss L(θ) parameter set θ in (4) is to perform the stochastic gradient descent (SGD) algorithm initialized with random gradient θ [29]. Today, the latest algorithm with enhanced convergence is the Adam Optimizer for dynamically adjusting learning rate η [13]. The Adam algorithm is used for handling classification problems during the training process. Since the structure of AE needs optical fiber simulation channel, we considered that all methods can be called from the TensorFlow library [30].

### 2.3. The Design of the Adaboost Algorithm Receiver

As the receiver, since the waveform is generated by ANN, ordinary algorithms, such as FFE, can only be used to improve the ISI between blocks and cannot identify the waveform generated by ANN. The usual demodulation method is not applicable, so it is impossible to identify the waveform and demodulate the information carried by the waveform. Therefore, this paper uses the Adaboost algorithm as the receiver. In the receiver side, we use the Adaboost algorithm to combine the Decision Trees as the receiver. The Adaboost training method is shown in Figure 2. At the beginning, the weights of all Decision Trees are initialized for the same parameters. The common training method of the Adaboost algorithm is to average all the parameters, and then train each Decision Tree. The output weight of the *k*-th weak learner of the training set is:(5)$$\;D\left(k\right)=\left(w_{k1},w_{k2},w_{k3},\ldots ,w_{km}\right);\;w_{1i}=\frac{1}{m}\;i=1,2,3,\ldots ,m.$$

The error rate of classification problems is well understood and calculated. Since multivariate classification is a generalization of binary classification, assuming that we are a binary classification problem, the output is {0,1}, and then the weighted error rate of the k-th weak classifier $G_{k}\left(x\right)$ on the training set is
(6)$$e_{k}=P\left(G_{k}\left(x_{i}\right)\neq y_{i}\right)=\sum_{i=1}^{m}w_{ki}I\left(G_{k}\left(x_{i}\right)\neq y_{i}\right).$$

The Adaboost algorithm generally uses a single-layer Decision Tree as its weak classifier. The single-layer Decision Tree is the most simplified version of the Decision Tree, which has only one decision point. It can be seen from the above formula that, if the classification error rate $e_{k}$ is larger, the corresponding weak classifier weight coefficient $\alpha_{k}$ is smaller. In other words, a weak classifier with a small error rate has a larger weight coefficient. Especially, the reason why this weight coefficient formula is used will be discuss when we investigate the loss function optimization of the Adaboost algorithm. Assuming that the weight coefficient of the sample set of the *k*-th weak classifier is $D\left(k\right)=\left(w_{k1},w_{k2},w_{k3},\ldots ,w_{km}\right)$, the weight coefficient of the sample set of the corresponding *k* + 1 weak classifier is
(7)$$w_{k+1,i}=\frac{w_{ki}}{Z_{K}}\exp\left(-\frac{1}{2}log\frac{1-e_{k}}{e_{k}}y_{i}G_{k}\left(x_{i}\right)\right).$$

From the calculation formula of $w_{k+1,i}$, it can be seen that, if the *i*-th sample is classified incorrectly, $y_{i}G_{k}\left(x_{i}\right)<0$, which causes the weight of the sample to increase in the (*k* + 1)-th weak classifier. If the classification is correct, then the weight is reduced in the (*k* + 1)-th weak classifier. Only one weak classifier is trained for each iteration, and the trained weak classifier will be used in the next iteration. That is to say, in the N-th iteration, there are a total of N weak classifiers, of which (N − 1) weak classifiers are previously trained, and their various parameters are no longer changed. Among them, the relationship of the weak classifiers is that the N-th weak classifier is more likely to match the data that the other weak classifiers did not match, and the final classification output depends on the comprehensive effect of these N classifiers.

Adaboost classification uses a weighted voting method, and the final strong classifier is
(8)$$f\left(x\right)=sign\left(\sum_{k=1}^{K}\frac{1}{2}log\frac{1-e_{k}}{e_{k}}G_{k}\left(x\right)\right).$$

For the updated sample weight, the sample set weight coefficient of the *k* + 1 weak classifier is
(9)$$w_{k+1,i}=\frac{w_{ki}}{\sum_{i}^{m}w_{ki}\alpha_{k}^{1-e_{ki}}}\alpha_{k}^{1-\frac{\left|y_{i}-G_{k}\left(x_{i}\right)\right|}{E_{k}}}\;i=1,2,3,\ldots ,m.$$

Finally, we obtain a strong classifier:(10)$$f\left(x\right)=G_{k^{*}}\left(x\right).$$

Among them, $G_{k^{*}}\left(x\right)$ is the weak classifier corresponding to the serial number $k^{*}$ corresponding to the median value of all $ln\frac{1}{\alpha_{k}},\;k=1,\;2,\;3,\ldots ,K$. Similarly, all methods can be called from the Scikit-learn library [31].

The Adaboost algorithm is an iterative algorithm in which its core idea is to train different classifiers (weak classifiers) for the same training set, and then group these weak classifiers to form a stronger final classifier. The advantage of the Adaboost algorithm is that its classification accuracy [13] is often better than that of complex neural network, while reducing the amount of computation on the classification. The advantage of this method is that it does not need to analyze the huge data iteratively, but through a simple weak classifier instead of ANN, more accurately for one or several cases, so as to improve the degree of signal identification and reduce the bit error rate.

### 2.4. The Design of Optical Fiber Simulation Channel

Similar to Reference [17], we considered the optical unamplified IM/DD link, which is the preferred solution for many low-cost short-range applications. Fiber dispersion and the nonlinearity produced by square-law PD photoelectric conversion are the main limiting factors in this communication channel. Due to the dispersive linear fiber channel model, the influence of a nonlinearity effect on the optical fiber channel is ignored, which means that there will be only inter-symbol crosstalk from the front and back. Therefore, as shown in Figure 1, the communication system on such a channel requires processing the sequence. In this work, we assume that a channel model includes LPF at the transmitter and receiver to reflect current hardware limitations, such as digital-to-analog and analog-to-digital converters (DAC/ADC), Mach-Zehnder modulation (MZM), partial discharge, electrically amplified noise, and optical fiber transmission.

In the simulation channel of Figure 1, the LPF before channel is designed as a 33 MHz low pass filter, which reflected the bandwidth of AWG. Similarly, the LPF after channel is designed as a 33 MHz low pass filter which reflected the bandwidth of the oscilloscope.

The other simulation channel can be expressed as a formula:(11)$$r\left(t\right)={\left|\hat{h}\left\{\hat{g}\left\{x\left(t\right)+n_{DAC}\left(t\right)\right\}\right\}\right|}^{2}+n_{REC}\left(t\right)+n_{ADC}\left(t\right).$$

The x(t) in the formula means the waveform after the 33 MHz LPF, and we first considered the noise of DAC (detailed description in Section 2.3). All the noise we considered is modeled as a Gaussian noise model. $\hat{g}\left(x\right)$ is modeled as MZM, which is $\hat{g}\left(x\right)=\sin\left(x\right)$. MZM in the simulation is designed as a sine function which can express in the range of [−π/4, π/4]. The influence of chromatic dispersion in the optical fiber channel on the signal is $\hat{h}\left(x\right)$. The influence of chromatic dispersion is added only in the frequency field; therefore, the time domain signal is converted to the frequency domain and multiplied by a parameter $H\left(\omega ,z\right)=\exp\left(-j\pi \frac{\beta }{2}z\omega^{2}\right)$ to represent the effect of dispersion (β means Fiber dispersion parameter, z means distance length, and ω means the angular frequency). Modulo square calculation is expressed as a photoelectric square law conversion. Therefore, the optical fiber is modeled as an intermediate medium that only provides attenuation and dispersion.

The noise from the DAC/ADC ($n_{DAC}\left(t\right)$ and $n_{ADC}\left(t\right)$) is modeled as additive and uniformly distributed, which is determined by ENOB (define it as 4). According to the experimental results, we estimate the variance $\sigma_{r}^{2}$ of the additive white Gaussian noise $n_{REC}\left(t\right)$. As we refer to experiment result, in order to better fit the actual situation, we adjust the Gaussian white noise amplitude to fit different distances. Our channel model includes fiber attenuation; thus, $\sigma_{r}^{2}$ is changeable as a function of the transmission distance. We apply the value of $\sigma_{r}^{2}$ = $1.564\times 10^{-3}$ for the examined systems of 25 km. Referring to Reference [17], we made some modifications to the formula as follows:(12)$$\sigma_{r}^{2}=3P\cdot 10^{-\frac{6.02\cdot ENOB+1.76}{10}}.$$

As the fiber model includes attenuation (0.17 dB/km), this will yield different effective SNRs at each transmission distance.

At the same time, due to the influence of dispersion on the channel, the whole system, with the increase of transmission distance, dispersion will seriously affect the bit error rate, so this paper focuses on the long-distance fixed length channel characteristic learning.

## 3. Experimental Verifications and Discussions

### 3.1. The Parameter and Method of System Training

For deep learning, the training method is significant. The training in simulation is similar to that in Reference [17]. Before experiments on optical fiber, we performed an autoencoder with symmetrical neural networks in the simulation on the optical fiber system in the simulation (Figure 1a). The parameters used in the simulation are discussed in the following table. As shown in Figure 1, as in Reference [17], ANN used in the receiver is symmetrical with ANN used in the transmitter in simulation, so as to train the transmitter to adapt optical fiber channel model. The design of the simulation channel is mentioned in Reference [17].

The process of simulation training is as follows: Binary Random Sequence created by Python is encoded into one-hot vector. One-hot vector is transmitted from the transmitter to the receiver across the simulation channel. After the ANN at the receiver, due to SoftMax, the output vector is probability distribution. The element with the greatest probability is considered to be the element most likely to be 1. Other elements are 0. Based on the concept of the autoencoder [13], the output vector after the autoencoder is equal to the input vector. Therefore, error emerges if the input vector is not equal to the output vector.

In order to train the transmitter, we use the aforementioned loss function (Formula (3)) and Adam optimizer to reduce the loss. After training 10,000 steps, we test BER by verification set and observe the loss whether is less than 1 × 10^−3^. If BER meets expectations, as in Reference [17], the transmitter is trained well.

The whole procedure of training is as follows: After training a symmetrical autoencoder with neural network, we send the waveform through the neural network on the transmitter and the optical fiber channel, and then we send it to the improved transmitter of the Adaboost algorithm (Figure 1b) for training. During the experiment, the input vector and labels must correspond one-to-one. The simulation parameters is below, see Table 1.

As an important point, the training process needs to set up mini-batch. Mini batch is usually the bigger, the better. The larger the batch size can be contained in the neural network and used as a training process, the better the training of classification accuracy is because it reduces the possibility of overfitting. Mini-batch is often the bigger, the better, but, after our training results, we found that the result is not the case. We tried to change the size of the mini-batch and obtain results. In the case of the different batch-size, we can see that, in fact, this will cause more changes in the loss value (the reason is that the more the data is put in, similarly is the loss doubled). The loss function is often very unstable when the mini-batch is bigger, and the fluctuation range of the loss function will be very large. The reason is that, when the matrix size of the input vector becomes larger, more changes will be caused. If the matrix size of the signal is not large enough, the loss is unable to converge under the same steps. It has a premise that the size of mini-batch includes all the possibilities of the information. Otherwise, the training model will cause an overfitting problem. However, the neural network is extremely sensitive to the regular changes of the input vector, but the unstable loss function caused by the large matrix size of your data should be avoided. We can select the batch-size which is suitable for the data size. Usually, choosing the batch-size close to the length of the signal is much more suitable than other sizes.

### 3.2. Experimental Setup and Results Discussions

The experimental block diagram is shown in Figure 3. First, we need to generate a random code instead of PRBS. Note that PRBS [32,33] cannot be the sequence generator because the neural network will overestimate the BER performance [34]. Before entering the neural network, in order to facilitate the neural network training, we map it into one-hot vector. The waveform mapped by the neural network is upsampled by N times, then sent to the LPF, and then sent to the RFA after DAC conversion to obtain the amplified waveform. Before entering the MZM, the MZM needs to be adjusted to the linear working area, usually set to be in the middle, between the highest voltage and the lowest voltage, due to the MZM instrument, and the optical power usually has a certain deviation. The input optical power is 5 dBm, and the output optical power will hover around −7 to −8 dBm. The fiber channel uses SSMF. As the transmission distance increases up to 55 km, adding the Erbium Doped Fiber Application Amplifier (EDFA) in the middle of the channel is necessary. Thermal noise caused by laboratory equipment has a strong impact on the waveform at the distance of more than 55 km. Although EDFA introduces amplifier spontaneous emission (ASE) noises, resulting in an Optical Signal Noise Ratio (OSNR) decrease, it amplifies the attenuated signal so that output symbols are able to be recognized by the neural network or Adaboost algorithm. The process of DSP is as follows. First, we perform cosine filtering, time recovery, dispersion compensation, and frame synchronization on the output waveform, so that the obtained waveform can correspond to the input of the transmitter one by one. Then, the Adaboost algorithm block enters for training. The method used for bit error rate is consistent with Figure 4. Note that, in contrast to Reference [17], the tunable dispersion module is not used to control the dispersion offset. Instead, a dispersion compensation algorithm is used in the DSP to deal with this problem. After the optical signal is converted into an electrical signal through PIN+TIA at the output end, the final result is obtained through ADC conversion. Due to the limitation of laboratory equipment, the oscilloscope we used cannot reach the sampling rate of 84 Gb/s, and the output waveform needs to be resampled during output.

As shown in Figure 4, the signal before and after passing through the channel will be slightly distorted and offset due to the existence of the fiber dispersion. For ANN with the strict one-to-one correspondence, the offset has a serious impact on the training. Therefore, in order to solve this problem, instead of downsampling at the receiver, the acquired waveform is processed directly by selecting one sampling point from every 7 sampling points as downsampling at this time, so that the offset is relatively small on the DSP, and BER is improved. It can be seen that, although it has some losses, it can be acceptable. Although the signal received at the receiver is somewhat damaged, its waveform is still very clear. This is why we process the entire waveform instead of picking out n points.

Before deciding the parameters of the AdaBoost algorithm, we first consider the influence of the received optical power on the system. As shown in Figure 5, we choose to measure the optical power at 35 km to show its versatility. Since we have already debugged MZM in the linear working area, the only one that can change the optical power is the laser transmitter. We can see that, with the increase of optical power, the eye diagram gradually becomes clear. When the optical power is about −8 dBm (Figure 5c), the eye diagram can gradually be divided into two levels. It can be seen that the best effect can be obtained when the linear working area of MZM is about −6 dbm. At this time, the working zero point of MZM curve can be obtained when the received optical power is about −6 dbm, when the BER tends to be flat. Therefore, we choose −6 dBm. Because we use different transmission distance, we can only control the transmission optical power, so we can obtain the transmission optical power of 3.6 dBm.

Adaboost algorithm uses a set of Decision Trees to solve the problem, two of which are worth noting, depth and the number of Decision Trees. In Figure 6a, BER performance by the Adaboost algorithm is much better than other Machine Learning methods since the Adaboost algorithm uses weight. The weight coefficient is used to determine the proportion of the classifier to determine the waveform, which is more conducive to the determination of the waveform. As shown in Figure 6b,c, using the control variable method, first, consider the comparison of the impact of the number of Decision Trees. In order to better reflect the advantages of the Adaboost algorithm in weak classifier planning, we use more Tree class methods for training as the receiver to receive the signal. We use different methods, such as Random Forest, Extra Tree, and Voting Tree, to compare with the Adaboost algorithm. Voting Tree adopts the voting mechanism, through voting on Extra Tree, Random Forest, and Decision Tree, to select the best classification result to obtain the lowest BER. It is obvious that, with the increase of the number of Trees, the change of bit error rate will tend to be flat. In addition, above that, in the Adaboost algorithm, compared with other methods, does not show advantages because it will produce over fitting phenomenon in the training process, which leads to signal misjudgment. Other methods also have this problem. In order to solve this problem, the traditional method is to limit the number of leaf nodes and the threshold value of each layer of leaves, while, in the communication system, the characteristics of the signal are complex and uncertain There are no rules, so we usually make the depth of the Tree deeper to traverse all kinds of possible signal situations to reduce the occurrence of bit error rate. Next, we analyze how to find the optimal number of points. It can be seen that, as the number of Decision Trees increases, the bit error rate drops sharply when the number is less than 20. As the transmission distance increases, the critical point at which the bit error rate tends to level off gradually comes back. The reason is that the more Decision Trees there are, the more serious the over-fitting phenomenon will be, and the waveform cannot be correctly classified.

Therefore, the redundant Decision Tree has no positive help for the reduction of bit error rate. Figure 6b shows the number of points at which each transmission distance tends to be flat. Considering the computational complexity and the best point for data, 30 is selected as the Number of Decision Trees point.

Then, we consider the depth of the Decision Tree. As shown in Figure 7a, it can be seen that, with the increase of depth alone, there is still a great gap between other Machine Learning methods and the Adaboost algorithm, and the decline of the Adaboost algorithm tends to moderate at last because it has reduced the bit error rate to the lowest possibility. As the depth of Tree increases, the repetitiveness of the waveform decision is easier to solve, and the decision distinction becomes more effective. Therefore, it can be said that the Adaboost algorithm achieves the optimal solution at the same distance. Next, we analyze how to find the optimal depth point. As shown in Figure 7b, it can be seen that the depth has a greater impact on the bit error rate relative to the number of Decision Trees. The depth of the Decision Tree will also tend to be flat when it reaches a certain depth. The reason is that, as the depth increases, it tends to be too sensitive to the analysis of some small changes in the signal waveform, which is actually harmful to the signal classification. After training, the classifier often knows that there are actually not two signal waveforms. This not only causes misjudgment but also increases training resources and computation-al complexity, which is of no benefit to reducing computational complexity. After the comparison of the aforementioned, Figure 7b, the depth of the short distance to the middle distance is often concentrated between 4 and 6. The following figure will continue to compare.

As shown in Figure 8, we first analyze the advantages and disadvantages between the neural network and Adaboost algorithm. For an autoencoder using a symmetrical neural network, it can be seen that it can still be lower than the adopted HD-FEC threshold for a long distance, and the output is relatively stable between 35 km and 65 km, but, at the distance of 75 km, the bit error rate will rise sharply because the inter-symbol interference caused by 75 km dispersion becomes more and more obvious, and the noise brought by the machine itself has gradually been able to compete with the power of its own signal. These are the reasons for the linear increase in the bit error rate. In addition, due to the usage of EDFA that amplifies the attenuated signal at more than 55 km, the output of 35–65 km is relatively stable as a result. Since the optical power emitted by the MZM has reached −7 to −8 dBm, the optical power obtained after 45 km has been lost to nearly −20 dBm. At this point, although neural networks and Decision Trees can still recognize symbols, they are already at the limit. Then, we analyze and compare the depth of the Adaboost algorithm Decision Tree. In the case of the same number of Decision Trees, it is obvious that the depth of the Tree will have a positive impact on data classification. However, too much depth is not our expectation. Due to limited experimental equipment, we cannot collect data to verify the performance of BER at different distances in order to analyze its robustness. We should be able to conclude that the Adaboost algorithm will be a certain degree of robustness at different distances. The reason is that, due to the relatively large number of Decision Trees, the Adaboost algorithm has a certain degree of robustness at different distances. However, compared to ANN, the figure of the BER performance will appear steeper in more changeable distance. Instead, compared to BiLSTM [18], the BER performance will be much poorer due to having none of the memory function. At the cost of excellent BER performance, the computation is more complex than the method we proposed. At this time, the weighted decision of multiple Decision Trees is obviously problematic. We will conduct experimental research on this in our future work.

Secondly, for situations where the possibility is relatively small, we do not need to increase the depth to 10 to reach the traversal level in all cases. When the depth is set to 5, the bit error rate is similar to the neural network, and, at the same time, it is close to the HD-FEC threshold at about 65 km. When the depth is 5, we can see in Figure 8b that, under the same conditions, the BER of other methods is nearly one order of magnitude lower than that of AdaBoost. Although the other methods are simpler in training and calculation than the AdaBoost algorithm, they sacrifice too much BER performance. Therefore, we just compare ANN with the AdaBoost algorithm. Due to similar BER performance at 65 km, the comparison between ANN and the Adaboost algorithm is relatively equal. Under this situation, comparison is meaningful. In this situation, the Adaboost algorithm is not very complicated in terms of computational complexity. A low bit error rate index is exchanged for complexity. The computational complexity analysis [4] of the neural network and Adaboost algorithm is as follows.

For the comparison method of computational complexity, we refer to Table 2 and Table 3. The biggest advantages of the Adaboost algorithm over neural networks are shown in the table above. It can be seen from Table 2 that, compared to neural networks, the Adaboost algorithm has two orders of magnitude lower multiplication operations. On the contrary, it only adds some addition and subtraction operations, which is a breakthrough in complexity. This is also a good foundation for real-time transmission of information. Since the autoencoder is equivalent to learning and adjusting the channel, for real-time transmission, channel adjustment will inevitably bring about re-training. At this time, the complexity of training must also be taken seriously. Obviously, as shown in Table 2 (the algorithm complexity of Adam and the Adaboost weight adjustment is not considered here, and the complexity of Adam is much higher), the training also transfers most of the multiplication operations to the multiplication operations, which is beneficial to the complexity. The reduction in complexity paves the way for rapid learning of changes in channel conditions later.

The above, Table 4 shows the speed difference between the two methods shown in ANN and Adaboost training and running. We use i7 7700 and Nvidia 2070 8 GB to train our system. Under the same conditions, each takes 100,000 one-hot vectors for training. Theoretically speaking, the ANN method should be more effective due to the use of TensorFlow-GPU [35] (the same problem is usually solved by GPU faster, and the general acceleration ratio is 3 to 10 times). There is no doubt that TensorFlow on the GPU definitely is achieved parallel by mini-batch in training. The only disadvantage of this algorithm is that it is transmitted to each other through PCIE during transportation so that a lot of time is spent on information, but, for complex operations, especially when FFT and other operations are added, we can often ignore the time spent on transportation. As far as the results are concerned, the Adaboost algorithm is obviously better than ANN in training and running time. The reason is that it saves a lot of multiplication operations. It is reflected in the fact that the equipment rarely uses multipliers, making the algorithm more efficient.

## 4. Conclusions

In this paper, an improved end-to-end autoencoder based on reinforcement learning by using Decision Tree for optical transceivers is proposed and experimentally demonstrated. Transmitters and receivers are considered as an asymmetrical autoencoder, combining the deep neural network and Adaboost algorithm. Experimental results show that 48 Gb/s with 7% HD-FEC threshold under 65 km SSMF is achieved with the proposed scheme. Compared to the autoencoder based on a Fully Connected Neural Network, our algorithm uses addition operations instead of multiplication operations, which can reduce computational complexity from 10^8^ to 10^7^ in multiplication and 10^6^ to 10^8^ in addition on the training phase.

## Figures and Tables

**Figure 1 micromachines-13-00031-f001:**
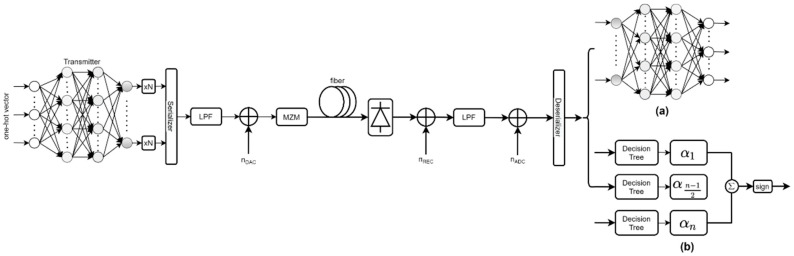
IM/DD optical fiber communication system with a deep Fully-Connected Neural Network combined with the receiver contains a Fully-Connected Neural Network (**a**) or the Adaboost algorithm (**b**) as the autoencoder.

**Figure 2 micromachines-13-00031-f002:**
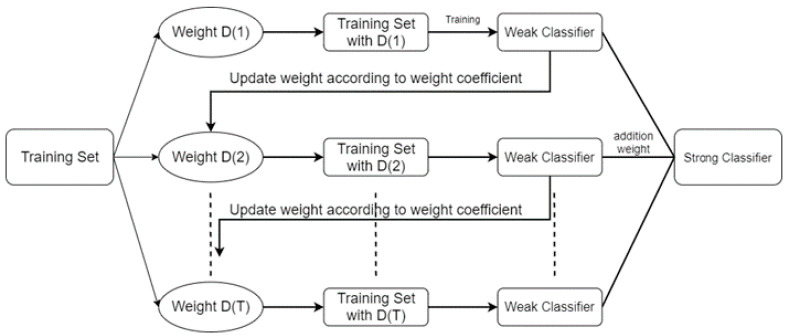
Training process of the Adaboost algorithm.

**Figure 3 micromachines-13-00031-f003:**
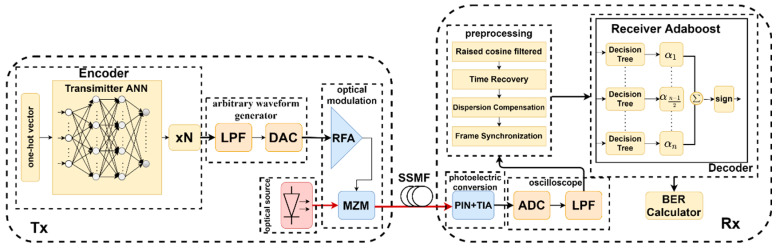
Training process of the Adaboost algorithm.

**Figure 4 micromachines-13-00031-f004:**
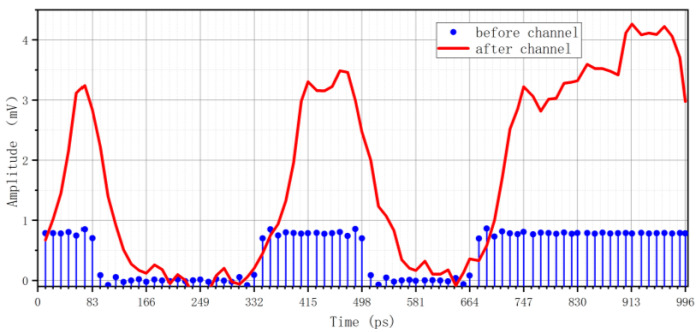
The influence of t the PD converted signal after passing through the channel.

**Figure 5 micromachines-13-00031-f005:**
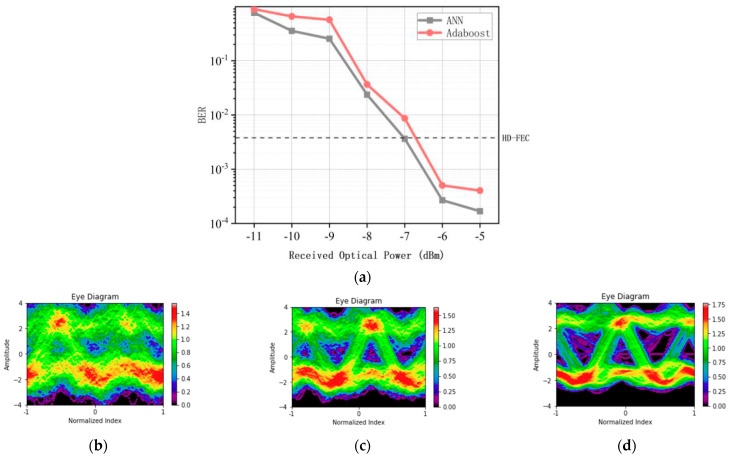
(**a**) The bit error rate performance of the Adaboost algorithm, ANN with received optical power in the distance of 35 km (the number of Adaboost Decision Tree is 30, and its depth is 5), and the eye diagram in the received optical power of −11 dBm (**b**), −8 dBm (**c**), −6 dBm (**d**).

**Figure 6 micromachines-13-00031-f006:**
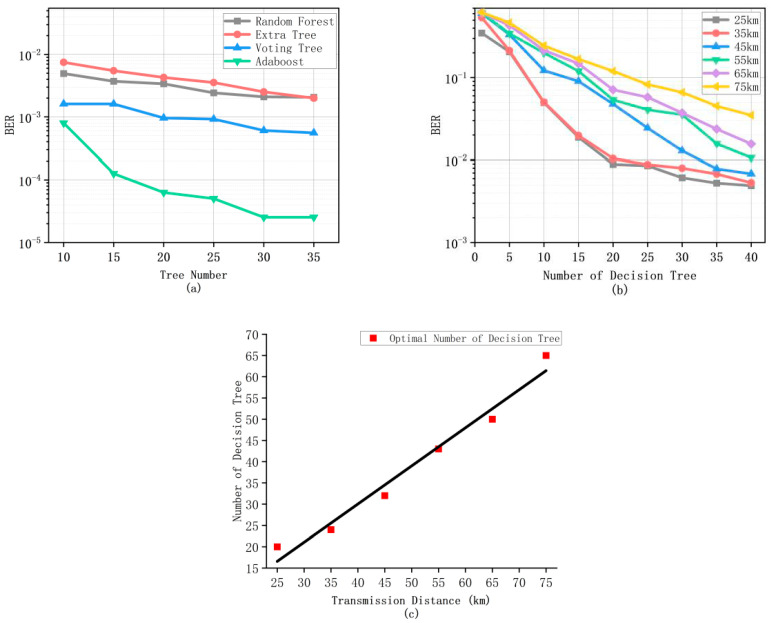
The bit error rate performance of the Adaboost algorithm, Random Forest, Extra Tree, and Voting Tree with the number of Decision Trees (**a**) and the Adaboost algorithm with the number of Decision Trees between 25 km to 75 km (**b**) and the relationship of the Adaboost algorithm with distance and its number (**c**).

**Figure 7 micromachines-13-00031-f007:**
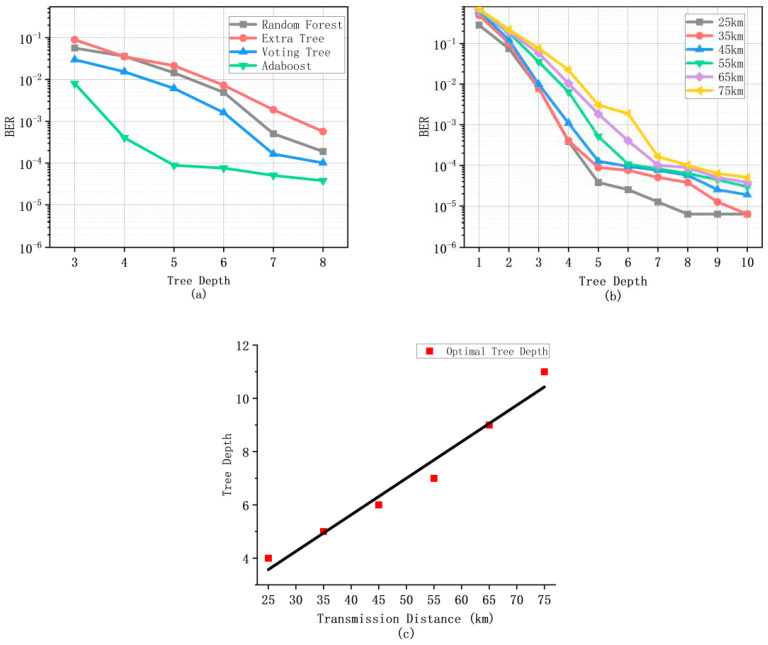
The bit error rate performance of Adaboost, Random Forest, Extra Tree, Voting Tree with the depth of Decision Trees (**a**) and the Adaboost algorithm with the depth of Decision Trees between 25 km to 75 km (**b**) and bit error rate performance of the Adaboost algorithm with its depth (km) (**c**).

**Figure 8 micromachines-13-00031-f008:**
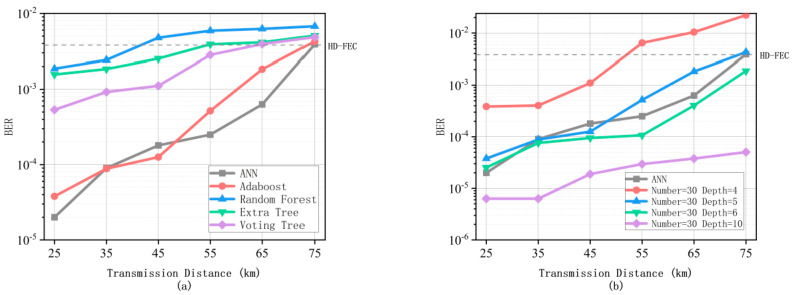
The bit error rate performance of Adaboost between different depth of Decision Tree and ANN (**a**) and the bit error rate performance of Adaboost compared with different Trees with the same situation (**b**) (Number = 30, Depth = 5).

**Table 1 micromachines-13-00031-t001:** Simulation parameters.

Parameter	Value
M	16
n	12
Sampling rate	84 GSa/s
Symbol rate	12 GSa/s
Information rate	4 bit/symbol
LPF bandwidth	33 GHz
DAC/ADC ENOB	4
Fiber dispersion parameter	17 ps/nm/km

**Table 2 micromachines-13-00031-t002:** The complexity comparison between ANN and the Adaboost classifier.

Algorithm	Complexity	$\mathrm{Calculation}\;(\times /+/x^{y})$
ANN	$C_{ANN}=\left(n_{i}n_{hid2}+n_{hid1}n_{hid2}+n_{out}\right)+\left(n_{out}\right)+\left(n_{out}\right)\;$	$C_{ANN}=\left(84\times 32+32\times 32+16\right)+16+16=3728+16+16$
Adaboost Classifier	$C_{Adaboost}=n_{n}+n_{n}\times \log_{2}n_{out}$	$C_{Adaboost}=30+120\;\;\;\;\;\;\;\;\;\;\;$

**Table 3 micromachines-13-00031-t003:** The training complexity comparison between ANN and the Adaboost classifier (20,000 steps).

Algorithm	Complexity	$\mathrm{Calculation}\;(\times /+/x^{y})$
ANN	$C_{ANN}=n\left[\left(n_{i}n_{hid2}+n_{hid1}n_{hid2}+n_{out}\right)+\left(n_{out}\right)+\left(n_{out}\right)\right]\;$	$C_{ANN}=32\times 10^{4}\times \left[\left(84\times 32+32\times 32+16\right)+16+16\right]=1.193\times 10^{8}+5.12\times 10^{6}+5.12\times 10^{6}\;$
Adaboost Classifier	$C_{Adaboost}=n\left[n_{n}+n_{n}\times \log_{2}n_{out}\right]$	$C_{Adaboost}=9.6\times 10^{6}+3.84\times 10^{8}$

**Table 4 micromachines-13-00031-t004:** The complexity comparison between ANN and the Adaboost classifier.

Algorithm	Training	Running
ANN(CPU)	287.24 s	259.64 s
ANN(GPU)	36.94 s	143.13 s
Adaboost Classifier	30.27 s	7.79 s

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
