# Peer review of "An Improved End-to-End Autoencoder Based on Reinforcement Learning by Using Decision Tree for Optical Transceivers"

_micromachines, 2021, doi:10.3390/mi13010031_

Round 1

Reviewer 1 Report

The paper discussed the end-to-end autoencoder in IM/DD systems using the decision tree with the Adaboost algorithm. The 48 Gb/s 68 km transmission was demonstrated using the proposed algorithm. However, the paper was not well written, and I prefer to NOT support its publications in Micromachines. Detail comments are as follows:

  1. Introduction. The introduction was not clear to some extent. For example, the authors used the reinforcement learning algorithm in the title. However, there were no descriptions and citations about reinforcement learning for End-to-end autoencoders. Did the Adaboost refer to the reinforcement learning here? In addition, please clarify the deep learning methods if the paper meant the end-to-end autoencoder. The authors preferred to use the word accuracy to describe the performance of the algorithms. What did the accuracy mean? The work focused on the comparison with [12], which used the BiLSTM, not FFNN, by the way. However, the author ignored the progress of the autoencoder for other works, such as the geometric shaping scheme. The other progress should be cited in the introduction. In terms of performance, the paper indicated the proposed algorithm outperformed the other algorithms in complexity and efficiency. But the author mentioned the improvement stem from sacrificing a small part of the performance in the second part. How much did the performance enhance? Please make the introduction more clear.
  2. Innovations. The paper used the Adaboost algorithms to equalize the inter-symbol interference between the symbols. I am confused why not use a simple FFE to compensate for the relevance between blocks. If so, what was the main innovation for the Adaboost algorithm? In addition, the results should compare the proposed method with the traditional DSP methods. Moreover, the autoencoder based on bits shows better performances than the one-hot vectors (Cammerer, Sebastian, et al. "Trainable communication systems: Concepts and prototype." IEEE Transactions on Communications 68.9 (2020): 5489-5503.). Why did the authors not use the bit inputs and outputs? Please comment on these opinions.
  3. Clarity. There were lots of unclear descriptions in the paper. The second part reviewed the autoencoder structures and introduced the proposed algorithm. In my opinion, it was more suitable and clear to separate the contents into two parts. In addition, Fig.1 showed the structure of the IM/DD system based on the proposed autoencoder. However, the receiver was still an FFNN in the figure. Please change the receiver with the decision tree combined with the Adaboost algorithm. In line 144, page 4, what did the signal feature mean? In line 247, page 6, what did the structure of autoencoder mean? Why did the structure of the autoencoder influence the channel characteristics? In line 272, page 7, why larger batch size can reduce the possibility of overfitting, and what was the batch size for training in the paper? In my opinion, the batch size influences the power normalization in the simulation, and a small batch size would be unstable for power normalization. How did the authors process the power normalization? Please explain in more detail.
  4. Please use the mathematical symbols in italics in the paper and improve the English writing skill and check the grammar. For example, in line 322, page 8, The should be lowercase.
  5. Results. Please clarify the generation method of the random sequence. The EDFA amplifies the attenuated signal and usually introduces the ASE noises resulting in the OSNR decrease instead of the SNR increase. In addition, the added EDFA should be shown in the figure. Why are the signals after the channel in good order in Fig 4.? Were the legends of signals reversed? The comparison between the ANN and Adaboost was unequal. What will performance be at the same complexity? In other words, How much will the complexity be at the same performance? In addition, [11] showed bad generalization performance for different distances. What will the Adaboost be when generalizing at a different distance after a fixed distance training?

Author Response

Authors’ Response to Reviewers

Title: An Improved End-to-end Autoencoder Based on Reinforcement learning By using Decision Tree for Optical Transceivers

Authors: Qianwu Zhang , Zicong Wang , Shuaihang Duan , Bingyao Cao , Yating Wu , Jian Chen , Hongbo Zhang , And Min Wang

Manuscript Number: Micromachine-1449251-2021

REVIEW

The paper discussed the end-to-end autoencoder in IM/DD systems using the decision tree with the Adaboost algorithm. The 48 Gb/s 68 km transmission was demonstrated using the proposed algorithm. However, the paper was not well written, and I prefer to NOT support its publications in Micromachines. Detail comments are as follows:

  1. The introduction was not clear to some extent. For example, the authors used the reinforcement learning algorithm in the title. However, there were no descriptions and citations about reinforcement learning for End-to-end autoencoders. Did the Adaboost refer to the reinforcement learning here? In addition, please clarify the deep learning methods if the paper meant the end-to-end autoencoder.

Response and Action Taken: Thanks very much for reviewing our paper and precious comments. AdaBoost is a method of reinforcement learning. In the revised version of the paper, the description of end-to-end communication system related to introduction and the relationship between reinforcement learning and AdaBoost algorithm are revised from (Section 1 paragraph6-7) Line84-Line103.

  1. The authors preferred to use the word accuracy to describe the performance of the algorithms. What did the accuracy mean?

Response and Action Taken:

Use accuracy in the following two sentences:

1.The advantage of the Adaboost algorithm is that its classification accuracy is often better than that of complex neural network while reducing the amount of computation on the classification. (Section 2.3 paragraph 7)

  1. The larger the batch can be contained in the neural network and used as a training process, the better the training of classification accuracy is, because it reduces the possibility of overfitting. (Section 3.1 paragraph 4)

Machine learning and deep learning usually have two models, one is regression model and the other is classification model. The Adaboost algorithm that we use is a classification model, which classifies and decodes the waveform of the receiver. If the output of the receiver is the same as the one-hot vector of the transmitter, we regard it as the correct classification. Otherwise, it is a wrong classification. Among them, the expression of accuracy means that AdaBoost algorithm expresses the accuracy of classification, and the commonly used expression of accuracy is the accuracy of classification.

In the revised version of MS, corresponding descriptions are added according to the reviewer’s comments.

  1. The work focused on the comparison with [12], which used the BiLSTM, not FFNN, by the way.

Response and Action Taken:

The quotation is wrong. We want to make comparison with [17]. In the revised version of MS, the corresponding description is corrected.

  1. However, the author ignored the progress of the autoencoder for other works, such as the geometric shaping scheme. The other progress should be cited in the introduction.

Response and Action Taken:

In the revised version of MS, the corresponding description is corrected. The introduction part of the paper really lacks the application and development of autoencoder in optical fiber communication. The paper has added the related introduction in line41-line83 (Section 1 paragraph 2-5).

  1. In terms of performance, the paper indicated the proposed algorithm outperformed the other algorithms in complexity and efficiency. But the author mentioned the improvement stem from sacrificing a small part of the performance in the second part. How much did the performance enhance? Please make the introduction more clear (clearer).

Response and Action Taken:

We consider to optimize the end-to-end design by replacing ANN with Adaboost algorithm on the receiver to reduce the computation complexity at the cost of sacrificing BER performance of the whole optical fiber system. Detailed analysis is added in Section 3 according to reviewer’s comments.

The optimization direction of the system is explained in detail. Because it cannot be quantified in the introduction, one is the computational complexity, and the other is to sacrifice some BER performance.

  1. The paper used the Adaboost algorithms to equalize the inter-symbol interference between the symbols. I am confused why not use a simple FFE to compensate for the relevance between blocks. If so, what was the main innovation for the Adaboost algorithm?

Response and Action Taken:

The premise of FFE is that also need to use neural network or other machine learning algorithms as the receiver. Otherwise, you will not be able to identify the information carried by the waveform. In theory, FFE can improve the ISI between the blocks, but you need to add a lot of computational complexity to complete the FFE design. For example, only 4-bit information used in this paper requires a lot of computational complexity. If the quantity of information becomes larger, it will not be worth the loss. In addition, in experiments, it is more difficult to achieve. If FFE is used in this paper, before adding it to the receiver, the training of the autoencoder is based on Tensorflow and needs to be trained as a whole graph. If complex FFE is added, the training time and actual operation will obviously outweigh the gains, so FFE is not used to process ISI between the blocks.

 In the revised version of MS, corresponding descriptions are added according to the reviewer’s comments. Detailed analysis is added in Section 2.3 according to reviewer’s comments.

  1. In addition, the results should compare the proposed method with the traditional DSP methods. Moreover, the autoencoder based on bits shows better performances than the one-hot vectors (Cammerer, Sebastian, et al. "Trainable communication systems: Concepts and prototype." IEEE Transactions on Communications 68.9 (2020): 5489-5503.). Why did the authors not use the bit inputs and outputs? Please comment on these opinions.

Response and Action Taken:

The core of this paper is to compare [17]. In [17], the traditional methods have been compared. Based on this, this paper restores his algorithm and makes an additional comparison.

In this paper, we use Python to generate binary random sequences. Obviously, we generate bit streams containing only 1 and 0. On this premise, we use one-hot coding by bit. It is no different from the article you mentioned.

Random code generated by Python Numpy Library is the bit stream that only consist of 0 and 1. Based on the bit stream, several bits represent the message m and is converted into one-hot encoding. The message m is encoded into a one-hot vector (containing "1" at position m and zero at other positions).

In the revised version of MS, corresponding descriptions are add according to the reviewer’s comments.

  1. There were lots of unclear descriptions in the paper. The second part reviewed the autoencoder structures and introduced the proposed algorithm. In my opinion, it was more suitable and clear to separate the contents into two parts.

Response and Action Taken: In the revised version of MS, corresponding descriptions are modified according to the reviewer’s comments

.

  1. In addition, Fig.1 showed the structure of the IM/DD system based on the proposed autoencoder. However, the receiver was still an FFNN in the figure. Please change the receiver with the decision tree combined with the Adaboost algorithm.

Response and Action Taken:

What is mentioned here is actually the structure of previous training of simulation channel model, which needs to be trained by using symmetric neural network similar to that mentioned in [17], and then the AdaBoost algorithm at the receiver is trained after training the transmitter.

In the revised version of MS, corresponding descriptions are modified according to the reviewer’s comments.

In Chapter 2.1, I posted the structure of the paper we recommended. This figure will continue to be used in training chapters in combination with relevant modifications to prevent misunderstanding.

  1. In line 144, page 4, what did the signal feature mean?

Response and Action Taken:

Since the signal phase modulated by the Mach Zehnder modulator is a linear region within [- π / 4, π / 4], it is necessary to consider outputting the signal into the above region.

In the revised version of MS, corresponding descriptions are modified according to the reviewer’s comments.

Due to Mach-Zehnder modulator (MZM) in IM/DD, unipolar signaling must be considered.

  1. In line 247, page 6, what did the structure of autoencoder mean? Why did the structure of the autoencoder influence the channel characteristics?

Response and Action Taken:

The paper refers to [17], in which the simulation channel is designed as a linear channel without considering the impact of nonlinearity, and the description in the original text is not very clear.

In the revised version of MS, corresponding descriptions are modified according to the reviewer’s comments.

Due to the dispersive linear fiber channel model, the influence of nonlinearity on the channel is not considered in the simulation.

  1. In line 272, page 7, why larger batch size can reduce the possibility of overfitting, and what was the batch size for training in the paper? In my opinion, the batch size influences the power normalization in the simulation, and a small batch size would be unstable for power normalization. How did the authors process the power normalization? Please explain in more detail.

Response and Action Taken:

Batch size refers to the number of one hot vectors at the same time at the input of neural network. Usually, too small will lead to slow convergence of the loss function and reduce the efficiency. Too small batch size will lead to failure of convergence of the loss function, failure of learning the channel model of the neural network and failure of training, Therefore, the number of tone-hot vector with similar specification to the input vector will be selected for training, so the training will be more effective and efficient. As for preventing overfitting, the reason is that without setting the batch size, there is too much difference in each individual waveform, which will reduce the gradient to a less ideal position, which is not conducive to our final result. Power normalization has nothing to do with the training optimization scheme. Power normalization is embodied in the receiver DSP, which is not the focus of this paper. In the revised version of MS, corresponding descriptions are added according to the reviewer’s comments. (Section 3.1 paragraph 4).

  1. Please use the mathematical symbols in italics in the paper and improve the English writing skill and check the grammar. For example, in line 322, page 8, The should be lowercase.

Response and Action Taken: We carefully examined the full text and improved our English writing according to the reviewer’s comments

.

  1. Please clarify the generation method of the random sequence.

Response and Action Taken: In the revised version of MS, corresponding descriptions are added according to the reviewer’s comments

(Section 2.1 paragraph 1):

Random Sequence generated by Python Numpy Library is the bit stream that only consist of 0 and 1.

  1. The EDFA amplifies the attenuated signal and usually introduces the ASE noises resulting in the OSNR decrease instead of the SNR increase. In addition, the added EDFA should be shown in the figure. Why are the signals after the channel in good order in Fig 4.? Were the legends of signals reversed?

Response and Action Taken:

Adding EDFA is to prevent excessive thermal noise from affecting the signal training without convergence at the receiver. The experiment shows that if it exceeds 55km, the signal at the receiving end cannot be trained and has no convergence due to excessive thermal noise. Therefore, EDFA amplification is adopted. the description in the original text is not very clear.

In the revised version of MS, corresponding descriptions are modified according to the reviewer’s comments. (Section 3.2 paragraph 1):

When reaching the distance of more than 55 km, adding EDFA in the middle of the channel is a must. Thermal noise caused by device cannot be ignored at the distance of more than 55km. Although the EDFA introduces the ASE noises resulting in the OSNR decrease, it amplifies the attenuated signal so that output symbols are able to be recognized by the neural network or Adaboost algorithm.

Fig 4. There is a problem with the annotation, which has been modified and indeed reversed. Thank you for pointing out.

  1. The comparison between the ANN and Adaboost was unequal. What will performance be at the same complexity? In other words, How much will the complexity be at the same performance?

Response and Action Taken:

The two algorithms cannot guarantee the same complexity, since one is machine learning algorithm, and the other is deep learning algorithm. The same performance is also difficult and can only be approximated. It can be seen in Fig. 8 (b) that the parameters mentioned by ANN and I have been approximated. On this premise, we compare the complexity of AdaBoost algorithm and the complexity of ANN, and come to the conclusion that the complexity of AdaBoost algorithm is much simple. In the revised version of MS, corresponding descriptions are added according to the reviewer’s comments.

  1. In addition, [11] showed bad generalization performance for different distances. What will the Adaboost be when generalizing at a different distance after a fixed distance training?

Response and Action Taken:

Thank you for asking this question. Due to the limitation of laboratory instruments, we cannot able to collect data to verify the performance of BER at different distances.

But according to the conclusions of [17], we should be able to conclude that the Adaboost algorithm will be a certain degree of adaptability at different distances. But compared to [17], the figure of the BER performance will appear steeper. We will conduct experimental research on this in our future work. The corresponding description is also added in the revised version. (Section 3.2 paragraph 6)

Reviewer 2 Report

The Authors present an alternative to typical Autoencoders architectures that is able to achieve significant complexity reduction at the same performance, which is of value for the promotion of the deployment of machine learning techniques in commercial transceivers. They adapt the classical structure of the symmetrical neural networks end-to-end systems, by replacing the receiver neural network by the Adaboost algorithm. While the Adaboost algorithm itself is not new, there is a novelty in the usage and study of this algorithm for the complexity reduction of autoencoders.

The document, however, is poorly structured, often using concepts that are only explained afterwards in the manuscript alongside with several non-trivial comments not justified nor referenced. Under these arguments, my advice is that the paper be accepted with major revisions, taking into consideration the following comments:

L57-59 -, It is not a wrong statement since reference [12] implements a recurrent neural network, and shows the gain of having memory in the transmitter and the receiver, and they compare it with memoryless feedforward neural networks.

L74 -75, In the following statement should be included a reference “In addition, multiple weak classifiers include various signal misjudgments caused by pulse broadening”

L88 “Adaboost algorithm obviously have stronger robustness” -> “Adaboost algorithm obviously has stronger robustness”

L103 “At the receiver,” -> “At the transmitter”

L107 The sentence “In order to reduce the complexity of selecting a weak classifier,” is confusing for the readers since “weak classifiers” are a specific term of Adaboost, and since this is part of the novelty on the work (to use Adaboost in an AE), these terms should not be used without first having been explained. Yet, Adaboost architecture is only explained afterward in the manuscript.

L121-123 repeated information, very confusing, should be rephrased.

L152 The authors should explain why “We have considered 7 times upsampling”.

L158 “R= log2M” Earlier in the paragraph, it is stated that each of the M messages is mapped to 7*n sampled symbols, which means that the IR is log2(M)/(7n) bits/symbol. Unless they mean "7n-sampled symbols", as in each symbol is constructed by 7n samples. Please clarify

L247-248, The authors should explain the reason “Due to the structure of autoencoder, the influence of nonlinearity on the channel is ignored in the simulation.”

L266-293 In Section 3.1, where the training of the transmitter is described, this information should be stated more clearly, namely what were the training parameters, and describe better the simulation channel, and what receiver was used for the symmetrical neural network during training;

L272 The sentence “Mini batch is usually the bigger the better.” is repeated again in L274-275

L289-293, The following sentence, “ After training a symmetrical neural…and labels must correspond one-to-one.” is an important part, it should be better explained since:

a) Previous lines don't sufficiently explain what they mean by asymmetrical NN (even though it is a bit obvious)

b) It is not clear what is the simulation channel model, the authors have said a few hints throughout previous paragraphs, but the information is sparse and confusing.

L296-298 The sentence “Note that PRBS [24] cannot be the code generator, the reason ..cannot estimate the method of the correct bit error rate.” I think that this is not true for memoryless neural networks. Since if there is no memory in decision, there is no capability of the network to reproduce PRBSs. This sentence should be better clarified and confirmed.

Author Response

Authors’ Response to Reviewers

Title: An Improved End-to-end Autoencoder Based on Reinforcement learning By using Decision Tree for Optical Transceivers

Authors: Qianwu Zhang , Zicong Wang , Shuaihang Duan , Bingyao Cao , Yating Wu , Jian Chen , Hongbo Zhang , And Min Wang

Manuscript Number: Micromachine-1449251-2021

REVIEW

The Authors present an alternative to typical Autoencoders architectures that is able to achieve significant complexity reduction at the same performance, which is of value for the promotion of the deployment of machine learning techniques in commercial transceivers. They adapt the classical structure of the symmetrical neural networks end-to-end systems, by replacing the receiver neural network by the Adaboost algorithm. While the Adaboost algorithm itself is not new, there is a novelty in the usage and study of this algorithm for the complexity reduction of autoencoders.

The document, however, is poorly structured, often using concepts that are only explained afterwards in the manuscript alongside with several non-trivial comments not justified nor referenced. Under these arguments, my advice is that the paper be accepted with major revisions, taking into consideration the following comments:

  1. L57-59 -, It is not a wrong statement since reference [12] implements a recurrent neural network, and shows the gain of having memory in the transmitter and the receiver, and they compare it with memoryless feedforward neural networks.

Response and Action Taken: Thanks very much for reviewing our paper and precious comments.

The quotation is wrong. We want to make comparison with [17]. In the revised version of MS, corresponding descriptions are modified according to the reviewer’s comments .

  1. L74 -75, In the following statement should be included a reference “In addition, multiple weak classifiers include various signal misjudgments caused by pulse broadening”

Response and Action Taken: Thanks very much for reviewing our paper and precious comments.

The reference is added [37]. This paper aims to mention that AdaBoost has a relatively good effect on classification. In the revised version of MS, corresponding descriptions are added according to the reviewer’s comments. (Section 1 paragraph 8)

  1. L88 “Adaboost algorithm obviously have stronger robustness” -> “Adaboost algorithm obviously has stronger robustness”

Response and Action Taken:

In the revised version of MS, corresponding descriptions are modified according to the reviewer’s comments. (Section 1 paragraph 8)

  1. L103 “At the receiver,” -> “At the transmitter”

Response and Action Taken: Thanks very much for reviewing our paper and precious comments.

We have modified the format of the whole paper to make it easier to understand. Here, we have modified it into an introduction to the receiver. (Section 2.1 paragraph 2-3)

  1. L107 The sentence “In order to reduce the complexity of selecting a weak classifier,” is confusing for the readers since “weak classifiers” are a specific term of Adaboost, and since this is part of the novelty on the work (to use Adaboost in an AE), these terms should not be used without first having been explained. Yet, Adaboost architecture is only explained afterward in the manuscript.

Response and Action Taken:

We adjust the position where the weak classifier appears. By modifying Figure 1, we simply describe the decision tree and weight coefficient used in Figure 1.

In the revised version of MS, corresponding descriptions are modified according to the reviewer’s comments (Section 2.1 paragraph 2-3):

In ANN (a), the ANN structure of the receiver is reversed with the ANN structure of transmitter. In Adaboost Algorithm (b), it is composed of several Decision Trees and its weight is updated by iterative training. Due to different training method, (a) can be trained together, and (b) cannot be trained together. Therefore, we need to train (b) after training (a) in order to get the ANN transmitter. (Detailed description in Section 3.1)

  1. L121-123 repeated information, very confusing, should be rephrased.

Response and Action Taken:

In the revised version of MS, corresponding descriptions are modified according to the reviewer’s comments.

(Section 2.2 paragraph 2):

Encoding method is as follow: The message m is encoded into a one-hot vector (the m-th element is equal to 1, and the other elements are 0) of size M, denoted as .

  1. L152 The authors should explain why “We have considered 7 times upsampling”.

Response and Action Taken:

Based on the limitation of laboratory instruments, the sampling rate of AWGN transmitted signal must be between 82Gsa/s and 92Gsa/s. if we choose the code rate of 12Gbit/symbol, we need 7 times up sampling to ensure that it is within this range. In the revised version of MS, corresponding descriptions are added according to the reviewer’s comments. (Section 2.2 paragraph 5)

  1. L158 “R= log2M” Earlier in the paragraph, it is stated that each of the M messages is mapped to 7*n sampled symbols, which means that the IR is log2(M)/(7n) bits/symbol. Unless they mean "7n-sampled symbols", as in each symbol is constructed by 7n samples. Please clarify

Response and Action Taken:

This refers to the conversion of an information m to a bit, and it is log2 (m) bit. Here, 7 times of up sampling represents the symbol rate. The description in the original text is not very clear.

In the revised version of MS, corresponding descriptions are modified according to the reviewer’s comments (Section 2.2 paragraph 5):

We have considered 7 times upsampling here in order to match the Arbitrary Wave-form Generator (AWG) sample rate.

  1. L247-248, The authors should explain the reason “Due to the structure of autoencoder, the influence of nonlinearity on the channel is ignored in the simulation.”

Response and Action Taken:

The paper refers to [17], in which the simulation channel is designed as a linear channel without considering the impact of nonlinearity, and the previous expression is inappropriate. Thank you for pointing out this problem.

In the revised version of MS, corresponding descriptions are modified according to the reviewer’s comments (Section 2.4 paragraph 1):

Due to the dispersive linear fiber channel model, the influence of nonlinearity on the channel is not considered in the simulation.

  1. L272 The sentence “Mini batch is usually the bigger the better.” is repeated again in L274-275

Response and Action Taken:

In the revised version of MS, corresponding descriptions are modified according to the reviewer’s comments (Section 3.1 paragraph 5).

  1. L289-293, The following sentence, “After training a symmetrical neural…and labels must correspond one-to-one.” is an important part, it should be better explained since:
  2. a) Previous lines don't sufficiently explain what they mean by asymmetrical NN (even though it is a bit obvious)
  3. b) It is not clear what is the simulation channel model, the authors have said a few hints throughout previous paragraphs, but the information is sparse and confusing.

Response and Action Taken:

In the revised version of MS, corresponding descriptions are modified according to the reviewer’s comments. The description of symmetric ANN is added (Section 3.1 paragraph 2-3). The simulated channel description has been added (Section 2.4 paragraph 2-4) and also mentioned in the reference paper [17], which is only cited due to the space limitation.

  1. L296-298 The sentence “Note that PRBS [24] cannot be the code generator, the reason .. cannot estimate the method of the correct bit error rate.” I think that this is not true for memoryless neural networks. Since if there is no memory in decision, there is no capability of the network to reproduce PRBSs. This sentence should be better clarified and confirmed.

Response and Action Taken:

This is the research on the application of PRBS to neural network equalizer in relevant paper [34].

[34] investigate the risk of overestimation when applying neural network in optical communication systems. It show that when using pseudo random bit sequences or short repeated sequences, the gain from applying neural network assisted receivers can be severely overestimated due to the capability of the NNs to learn to predict the pattern that is used.

In the paper, the neural networks used in the paper are the fully connected neural networks.By analyzing the simulation channel, this paper concludes that the neural network does not learn the principle of PRBS generation (Fully connected neural network does not have memory). However, the neural network will get the overestimation performance by long-term training neural network to predict the pattern that is used. The reason is that the generation of short sequence or PRBS will be regular, and the neural network will adjust its weight and bias (after all, different short sequences will have less stationaries. It is relatively simple and easy to be recognized, and the stationaries of PRBS may be relatively simple from the results) to reflect such regularity. Therefore, the use of neural networks will have a wrong overestimation of the training model in terms of the results. Therefore, in the research of autoencoder in this article, it is necessary to use random sequence to avoid this problem and prevent autoencoder from obtaining evaluation.

In the revised version of MS, corresponding descriptions are modified according to the reviewer’s comments. (Section 3.2 paragraph 1)

Round 2

Reviewer 1 Report

The authors have addressed most of the comments. However, the revised manuscript still has some problems to be addressed. First, the authors should modify the paper more carefully. For example, the sentence totally copied from the response letter should not be shown in the revised manuscript. In addition, the reviewer still has some comments/questions for the revised manuscript.

  1. The introduction mentioned the GAN and reinforcement learning for AE training. In my opinion, the main challenge of GAN is unstable training instead of complexity. Indeed, GAN can model the complex fiber channel at high accuracy with low complexity. It is well known that the training process of GAN is unstable and hard to converge. In addition, the mode collapse also is a serious problem for channel modeling. In terms of the model accuracy, GAN can be trained from the real channel, and the simulation model also is different from the actual channel. Thus, the description for GAN may be inaccurate. See the reference in detail.

Reference:

Yang, Hang, et al. "Fast and accurate optical fiber channel modeling using generative adversarial network." Journal of Lightwave Technology 39.5 (2020): 1322-1333.

  1. There were two versions of the manuscript in the PDF file. Were they the same? In addition, the response letter was WORD file with the revision mode. The sentence totally copied from the response letter should not be shown in the revised manuscript at line 481, page 14. Please pay more attention to the manuscript modification. In addition, please check the abbreviation. For example, there was no full name of SSMF in the paper.
  2. Please check the citations. In line 101, page 3, the citation [37] may be wrong. [37] is about the Pytorch instead of Adaboost algorithm. In addition, Numpy, TensorFlow, and Scikit Learn have published the journal paper, which can be cited in the manuscript. TensorFlow and Pytorch are two different packages. I think this work was based on TensorFlow. Why did the paper cite the Pytorch at line 517, pape 15?
  3. I am confused about parallel processing. In my opinion, the mini-batch training and processing on GPU are achieved parallel. Did the work process the ANN in serial processing? In addition, due to limited CMOS processing speed, a high-speed optical system usually demands parallelism.
  4. It is well known that the large batch size will result in the overfitting problem, showing bad generalization. See the references in detail. From my perspective, for the encoder training, the simulation needs power normalization before the channel. Small batch size leads to unstable normalization. But the author mentioned that power normalization has nothing to do with the training optimization scheme. How did the simulation process before the fiber channel?

Reference:

Keskar, Nitish Shirish, et al. "On large-batch training for deep learning: Generalization gap and sharp minima." arXiv preprint arXiv:1609.04836 (2016).

Hoffer, Elad, Itay Hubara, and Daniel Soudry. "Train longer, generalize better: closing the generalization gap in large batch training of neural networks." arXiv preprint arXiv:1705.08741 (2017).

  1. The performance generalization ability should be discussed in more detail. The authors clarified the Adaboost algorithm could be adapted to at different distances to some extent. Please give more details to explain it or support your comment. And the manuscript should compare briefly the features between the proposed mehtod and BiLSTM.

Author Response

Please refer to the word file for detailed cover letter.

Reviewer 2 Report

The authors performed well with all the corrections and explanations suggested. I have no further comments.

Author Response

Thank you for reviewing the manuscript. Your opinion is very important to the improvement of my paper.

Round 3

Reviewer 1 Report

The authors addressed my comments, and I believe the paper is in better shape now.  I would also encourage to authors to extend the work to improve the BER generalization performance for different transmission distances. The present reviewer thinks this revised manuscript could be accepted.

This manuscript is a resubmission of an earlier submission. The following is a list of the peer review reports and author responses from that submission.